# MoVieS: Motion-Aware 4D Dynamic View Synthesis in One Second

## Abstract

We present MoVieS, a novel feed-forward model that synthesizes 4D dynamic novel views from monocular videos in one second. MoVieS represents dynamic 3D scenes with pixel-aligned Gaussian primitives and explicitly supervises their time-varying motions. This allows, for the first time, the unified modeling of appearance, geometry and motion from monocular videos, and enables reconstruction, view synthesis and 3D point tracking within a single learning-based framework. By bridging view synthesis with geometry reconstruction, MoVieS enables large-scale training on diverse datasets with minimal dependence on task-specific supervision. As a result, it also naturally supports a wide range of zero-shot applications, such as scene flow estimation and moving object segmentation. Extensive experiments validate the effectiveness and efficiency of MoVieS across multiple tasks, achieving competitive performance while offering several orders of magnitude speedups. The code and pretrained model will be publicly available.

## 1 Introduction

Humans and animals perceive a continuous stream of observations from a dynamic 3D world, and effortlessly interpret its underlying geometry and motion. Replicating this capability is essential for any embodied agent that must understand and act in the physical world.

Recent advances have made great strides in individual 3D tasks, such as monocular depth estimation (Ranftl et al., 2020; Ke et al., 2024; Yang et al., 2024a; Bochkovskii et al., 2024; Chen et al., 2025a), 3D scene reconstruction (Schonberger & Frahm, 2016; Teed & Deng, 2021; Wang et al., 2024b; Zhang et al., 2025a; Wang et al., 2025a), novel view synthesis (Mildenhall et al., 2020; Fridovich-Keil et al., 2023; Kerbl et al., 2023; Wu et al., 2024; Luiten et al., 2024) and point tracking (Harley et al., 2022; Doersch et al., 2023; Wang et al., 2023; Karaev et al., 2024; Xiao et al., 2024). However, besides treating each task in isolation, most existing view synthesis and reconstruction studies focus on static scenes and require costly per-scene optimization without learning prior knowledge. Real-world environments are inherently dynamic and diverse, and all the aforementioned 3D scene understanding tasks could share common underlying principles.

Motivated by this, we introduce MoVieS, a **Mo**tion-aware dynamic **Vie**w **S**ynthesis model for feed-forward 4D reconstruction of monocular videos, that jointly models scenes' appearance, geometry and motion. MoVieS represents 3D dynamic scenes using renderable and deformable 3D particles, termed **dynamic splatter pixels**, and utilizes a differentiable 3D Gaussian rendering framework (Kerbl et al., 2023). Specifically, following recent practices on feed-forward view synthesis (Charatan et al., 2024; Szymanowicz et al., 2024; Zhang et al., 2024; Xu et al., 2025a; Lin et al., 2025a), each input pixel is mapped to a 3D Gaussian primitive, with its 3D location determined by predicted depth. To model dynamics, MoVieS regresses per-pixel motion displacements toward arbitrary query timestamps, enabling temporal tracking of each splatter pixel. This design facilitates coherent reconstruction of both 3D geometry and appearance across camera viewpoints and temporal frames.

As shown in Figure 1, MoVieS is build upon a large-scale pretrained transformer backbone (Oquab et al., 2024; Wang et al., 2025a), which encodes each video frame independently and aggregates their information via attentions (Vaswani et al., 2017). The aggregated features are then processed by specialized prediction heads: (1) a **depth head** estimates depth for each input frame, (2) a **splatter head** predicts per-pixel 3D Gaussian (Kerbl et al., 2023) appearance attributes, such as color and opacity, for novel view rendering, (3) a **motion head** estimates the time-conditioned movements of Gaussian primitives towards a target timestamp, allowing us to track its temporal evolution.

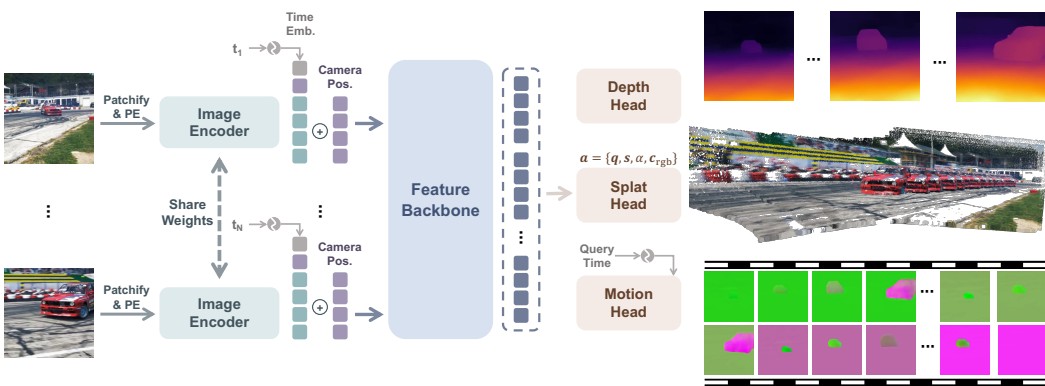

Figure 1: **Overview**. MOVIES consists of a shared image encoder, an attention-based feature backbone (Section 3.2.1), and three heads (Section 3.2.2) that simultaneously predict appearance, geometry and motion. Motion head is time-conditioned to model dynamic content with respect to several query timestamps. Time-varying Gaussian attributes are omitted for brevity.

Benefiting from the unified architecture, MOVIES can be trained on large-scale datasets featuring both static (Zhou et al., 2018; Wang et al., 2020; Li et al., 2023a) and dynamic (Mehl et al., 2023; Cabon et al., 2020) scenes, as well as point tracking datasets (Zheng et al., 2023; Karaev et al., 2023; Jin et al., 2025). During inference, it takes a monocular video, whether depicting a static or dynamic scene, and reconstructs per-pixel 3D Gaussian primitives along with their motion attributes at any target timestamp, enabling view synthesis, depth estimation, and 3D point tracking in a single model.

Extensive experiments on diverse benchmarks (Zhou et al., 2018; Yoon et al., 2020; Gao et al., 2022; Koppula et al., 2024) reveal that MOVIES achieves competitive performance across a variety of 4D perception tasks, while being several orders of magnitude faster than the existing state of the art. Furthermore, empowered by novel view synthesis as a proxy task, MOVIES enables dense motion understanding from sparse tracking supervision. This naturally gives rise to a variety of *zero-shot* applications, further broadening the potential of our approach.

In summary, our main contributions include:

- We introduce MOVIES, a novel feed-forward framework that jointly models appearance, geometry and motion for 4D scene perception from monocular videos.
- Dynamic splatter pixels are proposed to represent dynamic 3D scenes as renderable deforming 3D particles, bridging novel view synthesis and dynamic geometry reconstruction.
- MOVIES delivers strong performance and orders of magnitude speedups for 4D reconstruction, and naturally enables a wide range of applications in a zero-shot manner.

## 2 RELATED WORK

### 2.1 FEED-FORWARD 3D RECONSTRUCTION

Traditional 3D reconstruction relies on dense multi-view supervision and per-scene optimization (Hartley & Zisserman, 2003; Mur-Artal et al., 2015; Schonberger & Frahm, 2016; Mildenhall et al., 2020). Recent learning-based methods leverage large-scale priors to enable feed-forward prediction of depth, pose, and other 3D properties (Wang et al., 2024a; Ranftl et al., 2020; Piccinelli et al., 2024; Hu et al., 2024). With differentiable rendering such as 3D Gaussian Splatting (3DGS) (Kerbl et al., 2023), some works further regress pixel-aligned Gaussian attributes for novel view synthesis (Charatan et al., 2024; Chen et al., 2024; Zhang et al., 2024). DUSt3R (Wang et al., 2024b) pioneered direct regression of pixel-aligned pointmaps in a canonical space from image pairs. Its extensions (Leroy et al., 2024; Wang & Agapito, 2025; Liu et al., 2025b; Smart et al., 2024; Zhang et al., 2025b) generalize this framework with multi-view, streaming, and 3DGS integration. VGGT (Wang et al., 2025a) unifies these advances with a strong image encoder (Oquab et al., 2024) and task-specific heads, but all remain limited to static scenes. In contrast, we extend feed-forward 3DGS reconstruction to dynamic environments with moving objects.

## 2.2 DYNAMIC RECONSTRUCTION AND VIEW SYNTHESIS

Compared to static reconstruction, dynamic reconstruction remains underexplored. Extensions of DUSt3R (Chen et al., 2025b; Sucar et al., 2025; Jin et al., 2025) handle dynamic reconstruction in a plug-and-play manner (Chen et al., 2025b) or leveraging foundation models (Zhang et al., 2025a; Lu et al., 2025; Yao et al., 2025) with monocular depth (Bochkovskii et al., 2024; Piccinelli et al., 2025), optical flow (Teed & Deng, 2020), and point tracking (Karaev et al., 2024), but remain limited to two-frame inputs and output only sparse point clouds. CUT3R (Wang et al., 2025c) extends to video inputs via recurrent updates, while diffusion-based methods (Xu et al., 2025b; Jiang et al., 2025) treat reconstruction as conditional generation, though at the cost of multiple passes per sequence.

Dynamic novel view synthesis focuses on rendering quality and view-dependent effects, instead of geometric structure. NeRF-based methods (Mildenhall et al., 2020; Fridovich-Keil et al., 2023; Cao & Johnson, 2023) represent dynamic scenes from multi-view (Bansal et al., 2020; Wang et al., 2022; Li et al., 2022) or monocular videos (Pumarola et al., 2021; Park et al., 2021; Li et al., 2023b; Zhao et al., 2024), while 3DGS (Kerbl et al., 2023) extends this with 4D primitives (Yang et al., 2024b; Li et al., 2024) or deformable fields (Wu et al., 2024; Luiten et al., 2024; Yang et al., 2024c; Lin et al., 2024; Liu et al., 2025a). Recent works further express Gaussian motion explicitly (Wang et al., 2025b; Sun et al., 2024; Stearns et al., 2024; Lei et al., 2025; Lin et al., 2025b), but requiring scratch training, iterative optimization, and external supervision from point tracking or optical flow estimation (Doersch et al., 2023; Teed & Deng, 2020).

For feed-forward 4D view synthesis, STORM (Yang et al., 2025) is designed for outdoor driving scenes from multi-view videos. BTimer (Liang et al., 2024) and NutWorld (Shen et al., 2025) estimate 3DGS attributes from monocular inputs. However, BTimer predicts independent Gaussian chunks per timestamp without modeling frame relations and needs an enhancer module for smooth intermediate frames. NutWorld models Gaussian motion but lacks explicit supervision, relying heavily on pretrained depth (Chen et al., 2025a) and flow (Xu et al., 2023) estimation models, and uses an orthographic camera, which may further lead to projection distortions.

## 3 METHOD

### 3.1 DYNAMIC SPLATTER PIXEL

To model 3D scenes with moving contents, we propose a novel representation, namely **dynamic splatter pixel**, which decomposes dynamic scenes into a set of static Gaussian primitives and their corresponding deformation fields. Given an input video $\mathcal{V}$, each pixel in the $i$-th frame $\mathbf{I}_i$ is associated with a splatter pixel $\mathbf{g}_i$ (Charatan et al., 2024; Szymanowicz et al., 2024; Zhang et al., 2024) in a *shared* canonical space of the first frame camera's coordinate system. Each $\mathbf{g} \coloneqq \{\mathbf{x}, \mathbf{a}\}$ is parameterized by its position $\mathbf{x} \in \mathbb{R}^3$ in the canonical space and other rendering attributes $\mathbf{a} \in \mathbb{R}^{11}$, including rotation quaternion $\mathbf{q} \in \mathbb{R}^4$, scale $\mathbf{s} \in \mathbb{R}^3$, opacity $\alpha \in \mathbb{R}$, and color $\mathbf{c}_{\text{rgb}} \in \mathbb{R}^3$ (Kerbl et al., 2023). Considering splatter pixels were originally designed for static scenes, we decouple motion from geometric structure to adapt them for dynamic scenes. An additional time-dependent **deformation field** is introduced, in which each splatter pixel $\mathbf{g}$ is associated with $\mathbf{m}(t) \coloneqq \{\Delta\mathbf{x}(t), \Delta\mathbf{a}(t)\}$. $\Delta\mathbf{x}(t) \in \mathbb{R}^3$ is the motion vector of a splatter pixel at time $t$ with respect to the *canonical* 3D space, and $\Delta\mathbf{a}(t)$ is the change of the corresponding attributes of the splatter pixel at time $t$. Therefore, the splatter pixel $\mathbf{g}$ is deformed to time $t$ as:

$$\mathbf{x} \leftarrow \mathbf{x} + \Delta\mathbf{x}(t), \quad \mathbf{a} \leftarrow \mathbf{a} + \Delta\mathbf{a}(t). \tag{1}$$

By combining static splatter pixels and their deformation fields, we establish the correspondence between each Gaussian primitive and its temporal dynamics, thereby enabling dynamic scene modeling and dense motion estimation.

### 3.2 MOVIES: UNIFY APPEARANCE, GEOMETRY AND MOTION

#### 3.2.1 FEATURE BACKBONE

Given a posed video $\mathcal{V} = \{\mathbf{I}_i, \mathbf{P}_i, \mathbf{K}_i, t_i\}_{i=1}^N$, we first patchify each input image $\mathbf{I}_i$ and use a pretrained image encoder (Oquab et al., 2024) to extract their features as shown in Figure 1. To effectively incorporate camera information, we adopt two complementary strategies to embed the

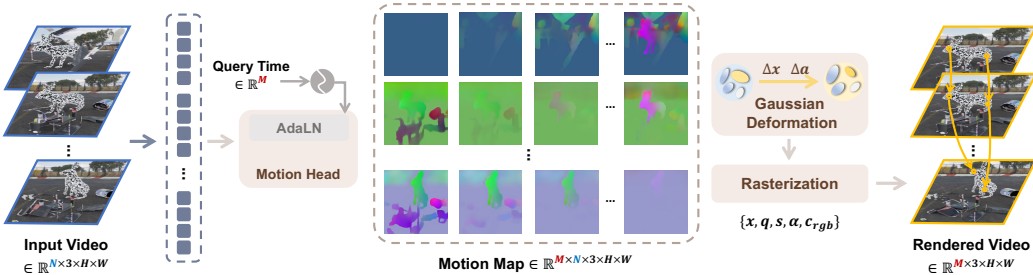

Figure 2: **Motion Head**. Given $M$ query timesteps, the proposed motion head is conditioned via adaptive layer normalization (AdaLN) and predicts 3D displacements for each input pixel. After rasterization using the $M$ corresponding query-time cameras, output images in shape $M \times 3 \times H \times W$ are rendered for supervision. Gaussian attribute deformation $\Delta \mathbf{a}$ is omitted for brevity.

camera parameters into image features: (1) **Plücker embedding**: camera pose $\mathbf{P}_i$ and intrinsics $\mathbf{K}_i$ are transformed into pixel-aligned Plücker embeddings (Sitzmann et al., 2021), which are then downsampled and fused with image features via spatial-wise addition; (2) **Camera token**: $\mathbf{P}_i$ and $\mathbf{K}_i$ are passed through a linear layer and encoded to a camera token, which is appended to the sequence of image tokens. Ablation study on these two camera injection manners is validated in Section 4.4.

To inform the model that input images originate from a temporally ordered video, we additionally encode the timestamp $t_i \in [0, 1]$ of input frames by sinusoidal positional encoding (Mildenhall et al., 2020) to produce a **timestamp token**, which is then concatenated with the aforementioned image and camera tokens as shown in Figure 1. After tokenizing input images, camera parameters, and timestamps, we apply geometrically pretrained attention blocks (Vaswani et al., 2017) from VGGT (Wang et al., 2025a) to enable interactions among image tokens across video frames. This produces a set of shared feature tokens enriched with inter-frame context as well as camera and temporal information, which are then used for predicting various properties of dynamic scenes.

### 3.2.2 PREDICTION HEADS

Aggregated tokens from the feature backbone are fed into three parallel prediction heads that estimate the appearance, geometry and motion of the dynamic scene respectively as shown in Figure 1. Each head adopts a DPT-style architecture (Ranftl et al., 2021) to convert feature tokens into dense predictions matching the input resolution, thereby producing dynamic splatter pixels.

**Depth and Splatter Head**   Different from previous feed-forward 3DGS reconstruction methods (Charatan et al., 2024; Chen et al., 2024; Zhang et al., 2024) that use a single head to predict all splatter pixel attributes, we adopt a decoupled design to better leverage geometric priors from the pretrained VGGT (Wang et al., 2025a). Specifically, a dedicated **depth head**, initialized from VGGT, is used for geometry prediction to provide spatial grounding for splatter pixel construction, while another DPT as **splatter head** is trained from scratch for appearance rendering. We further incorporate a direct RGB shortcut (Ye et al., 2025a) from the input image to the final convolution layer of the splatter head to preserve high-frequency details and enhance color fidelity.

**Motion Head**   To capture scene dynamics, a novel **motion head** as shown in Figure 2 is introduced to predict dense deformation $\mathbf{m}(t)$ for each dynamic splatter pixel at any target moment. Temporal variation is enabled by injecting the sinusoidally encoded query time $t_q$ into the aggregated tokens via adaptive layer normalization (Xu et al., 2019) before applying DPT convolutions. For each input frame $\mathbf{I}_i$ at time $t_i$, the motion head predicts its 3D movements $\Delta \mathbf{x}$ and Gaussian attribute deformation $\Delta \mathbf{a}$ toward $t_q$ in a *shared* world coordinate system.

## 3.3 TRAINING

### 3.3.1 DATASET CURATION

An ideal dataset for dynamic scene reconstruction would include synchronized multi-view videos with dense depth and point tracking annotations. However, such data is infeasible to capture and annotate at scale in practice. Instead, we leverage a diverse set of open-source datasets (Zhou et al.,

Table 1: **Training Datasets**. Eight datasets from diverse sources are utilized to train MOVIES at scale. "#Repeat" denotes dataset duplication count during integration to balance their contributions.

| Dataset | Dynamic? | Depth? | Tracking? | Real? | #Scenes | #Frames | #Repeat |
|---|---|---|---|---|---|---|---|
| RealEstate10K (Zhou et al., 2018) | ✗ | ✗ | ✗ | ✔ | 70K | 6.36M | 1× |
| TartanAir (Wang et al., 2020) | ✗ | ✔ | ✗ | ✗ | 0.4K | 0.49M | 100× |
| MatrixCity (Li et al., 2023a) | ✗ | ✔ | ✗ | ✗ | 4.5K | 0.31M | 10× |
| PointOdyssey (Zheng et al., 2023) | ✔ | ✔ | ✔ | ✗ | 0.1K | 0.18M | 1000× |
| DynamicReplica Karaev et al. (2023) | ✔ | ✔ | ✔ | ✗ | 0.5K | 0.26M | 100× |
| Spring (Mehl et al., 2023) | ✔ | ✔ | ✗ | ✗ | 0.03K | 0.003M | 2000× |
| VKITTI2 (Cabon et al., 2020) | ✔ | ✔ | ✗ | ✗ | 0.1K | 0.03M | 500× |
| Stereo4D (Jin et al., 2025) | ✔ | ✔ | ✔ | ✔ | 98K | 19.6M | 1× |

2018; Wang et al., 2020; Li et al., 2023a; Mehl et al., 2023; Cabon et al., 2020; Zheng et al., 2023; Karaev et al., 2023; Jin et al., 2025), each providing *complementary* supervision, as shown in Table 1. With the flexible model design, MOVIES can be trained on these heterogeneous sources by aligning objectives to their respective annotations. Details about data curation are provided in Appendix A.

### 3.3.2 OBJECTIVES

MOVIES is trained by a multi-task objective that combines depth, rendering and motion losses:

$$\mathcal{L} := \lambda_{\mathrm{d}}\mathcal{L}_{\mathrm{depth}} + \lambda_{\mathrm{r}}\mathcal{L}_{\mathrm{rendering}} + \lambda_{\mathrm{m}}\mathcal{L}_{\mathrm{motion}}. \tag{2}$$

**Depth and Rendering Losses** Depth loss is computed as the mean squared error (MSE) between the predicted and ground truth depth maps, along with their spatial gradients, after filtering out invalid values. Rendering loss combines pixel-wise MSE and perceptual loss (Zhang et al., 2018) between 3DGS-rendered images under corresponding camera views and the video frames at target timestamps, which are randomly sampled during training.

**Motion Loss** Given 3D point tracking datasets, ground-truth motion $\Delta\mathbf{x}$ is defined as the 3D displacement of each tracked point between any two frames. Since all 3D points are defined in the world coordinate and most tracked points remain static, implying that their corresponding motion vectors tend to zero. We apply a point-wise L1 loss between the predicted and ground-truth motions to promote sparsity after filtering out points that are not visible in the input frames. Additionally, to complement direct point-to-point alignment, a **distribution loss** is introduced that encourages the predicted motion vectors to preserve the internal relative distance structure within each frame. The final motion loss is defined as a combination of point-wise and distribution-level supervision:

$$\mathcal{L}_{\mathrm{motion}} := \lambda_{\mathrm{pt}}\mathcal{L}_{\mathrm{pt}} + \lambda_{\mathrm{dist}}\mathcal{L}_{\mathrm{dist}} \tag{3}$$

$$= \frac{1}{P}\sum_{i\in\Omega}\lambda_{\mathrm{pt}}\|\Delta\hat{\mathbf{x}}_i - \Delta\mathbf{x}_i\|_1 + \frac{1}{P^2}\sum_{(i,j)\in\Omega\times\Omega}\lambda_{\mathrm{dist}}\|\Delta\hat{\mathbf{x}}_i \cdot \Delta\hat{\mathbf{x}}_j^\top - \Delta\mathbf{x}_i \cdot \Delta\mathbf{x}_j^\top\|_1, \tag{4}$$

where $\Omega$ denotes the set of all valid $P$ tracked points, and $\Omega\times\Omega$ is its Cartesian product. Ablation study on the effectiveness of these two types of motion supervision is presented in Section 4.4.

**Normalization** Similar to VGGT (Wang et al., 2025a), we normalize the 3D scene scale by the average Euclidean distance from each 3D point to the origin of the canonical world coordinate system. As a result, unlike some other reconstruction methods (Wang et al., 2024b; Leroy et al., 2024; Wang et al., 2025c), we do not apply additional normalization in the depth or motion loss. We also omit confidence-aware weighting (Wang et al., 2024b; 2025a) for simplicity and more stable training.

## 4 EXPERIMENTS

### 4.1 EXPERIMENTAL SETTINGS

**Implementation** MOVIES is built on a geometrically pretrained transformer, VGGT (Wang et al., 2025a), with splatter head and camera/time embeddings trained from scratch. AdamW (Loshchilov & Hutter, 2018) with cosine learning rate scheduling and linear warm-up is used for optimization. We observed that training MOVIES is particularly unstable, likely arising from sparse annotations and the heterogeneous nature of the training data. A curriculum strategy is employed that gradually

increases the training complexity, involving (1) pretraining on static scenes, (2) dynamic scenes with varying views, and (3) fine-tuning on high resolution. Several techniques, such as gsplat rendering backend (Ye et al., 2025b), DeepSpeed (Rasley et al., 2020), gradient checkpointing (Chen et al., 2016), gradient accumulation, and bf16 mixed precision, are adopted to improve memory and computation efficiency. Training completes in approximately 5 days using 32 H20 GPUs. More details about implementations are provided in Appendix A.

**Evaluation** We evaluate the capabilities of MOVIES on two primary tasks: novel view synthesis (Section 4.2) and 3D point tracking (Section 4.3). Following prior works (Charatan et al., 2024; Chen et al., 2024; Xu et al., 2025a; Wang et al., 2025b; Sun et al., 2024; Lei et al., 2025; Liang et al., 2024), RealEstate10K (Zhou et al., 2018) is used to evaluate novel view synthesis performance on static scenes. DyCheck (Gao et al., 2022) and NVIDIA dynamic scene dataset (Yoon et al., 2020) are adopted for dynamic scenes, in terms of PSNR, SSIM and LPIPS (Zhang et al., 2018). Since the DyCheck dataset contains invisible regions in the target novel views, we compute reconstruction metrics using the provided covisibility masks (denoted by a prefix "m"), ensuring a fair comparison across methods. For 3D point tracking, the TAPVid-3D (Koppula et al., 2024) benchmark serves as the evaluation protocol, covering both indoor and outdoor scenes across three datasets. End-point Euclidean error in the 3D space ($EPE_{3D}$) and the percentage of 3D points within 0.05 and 0.10 units of the ground-truth locations ($\delta_{3D}^{0.05}$ and $\delta_{3D}^{0.10}$) are utilized as metrics.

## 4.2 NOVEL VIEW SYNTHESIS

### 4.2.1 STATIC SCENE VIEW SYNTHESIS

As a special case of dynamic scenes, we first evaluate MOVIES on a static dataset, RealEstate10K (Zhou et al., 2018), comparing against several state-of-the-art feed-forward static scene reconstruction methods, including DepthSplat (Xu et al., 2025a), our reimplementation of GS-LRM (Zhang et al., 2024), and our method pretrained solely on the first-stage static reconstruction. As shown in Table 2, although MOVIES is primarily designed for dynamic scenes, it maintains competitive performance on static scenes. Notably, when processing static inputs, our predicted motion naturally converges to *zero*, demonstrating the ability of MOVIES to implicitly differentiate between static and dynamic regions without explicit supervision.

### 4.2.2 DYNAMIC SCENE VIEW SYNTHESIS

For dynamic scene view synthesis, we compare against three state-of-the-art open-sourced methods (Wang et al., 2025b; Sun et al., 2024; Lei et al., 2025) for dynamic 3D Gaussian reconstruction, and also report static feed-forward baselines for reference. As shown in Table 2, MOVIES achieves competitive or superior performance compared to baselines, while requiring only 0.93s per scene, which is orders of magnitude faster than prior approaches that rely on heavy pretrained models and complex multi-stage pipelines. Qualitative visualizations in Figure 3 further highlight the strength of our approach. While MoSca (Lei et al., 2025) demonstrates impressive performance, it struggles with sparse inputs and often overfits to seen poses, producing spiky and over-saturated artifacts under novel views and timesteps. In contrast, MOVIES leverages large-scale learned priors to generalize more effectively, yielding smoother and more realistic results.

To ensure fair comparison and better reflect real scenarios, no video masks for dynamic objects are used in our experiments. It poses a major challenge for optimization-based methods like Shape-of-Motion (Wang et al., 2025b) and Splatter-a-Video (Sun et al., 2024), which rely heavily on explicit motion segmentation. The challenge is particularly evident on the NVIDIA dataset, where significant camera shake hinders disentangling scene dynamics, leading to notably degraded performance and even worse than static baselines. In contrast, MOVIES exhibits strong robustness by directly learning to model motion. More visualization results are provided in Figure 6 and the supplementary material.

## 4.3 3D POINT TRACKING

Trained on large-scale point tracking datasets (Zheng et al., 2023; Karaev et al., 2023; Jin et al., 2025), the proposed method can also *densely* track any 3D point corresponding to a pixel across video frames. We compare MOVIES against three strong baselines, including two state-of-the-art 2D point tracking methods, BootsTAP (Doersch et al., 2024) and CoTracker3 (Karaev et al., 2024), as well as a native 3D point tracking approach, SpatialTracker (Xiao et al., 2024). For 2D trackers, a recent video depth estimation model (Chen et al., 2025a) and ground-truth camera intrinsics are

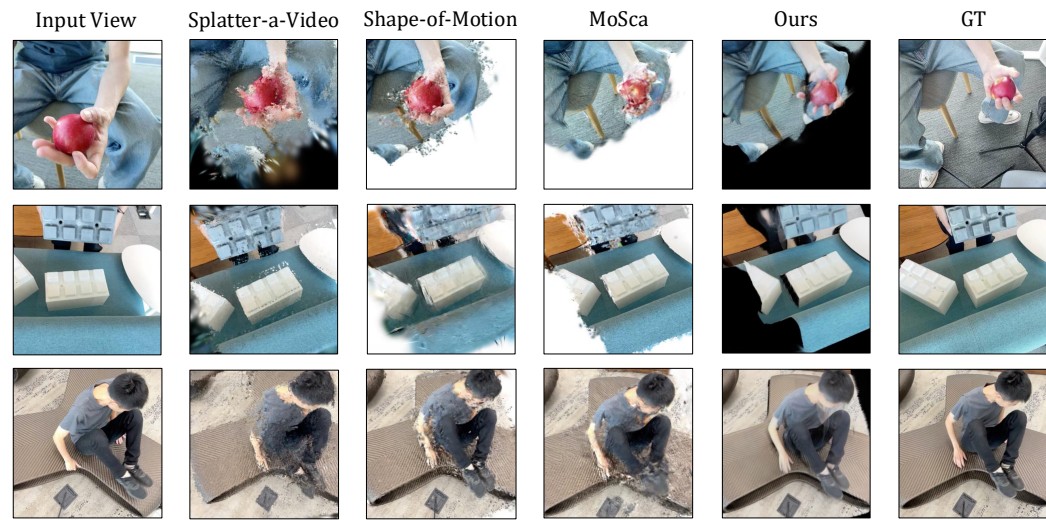

Figure 3: **Novel View Synthesis for Dynamic Scenes**. Given a monocular video, we compare synthesized views from a novel view across different methods. Regions invisible in the input are rendered as black or white, depending on the rendering implementation. More results in Figure 6.

Table 2: **Evaluation on Novel View Synthesis**. The best , second best and third best results are highlighted for clarity. † indicates our reimplemented version of GS-LRM (Zhang et al., 2024).

| Novel View Synthesis | Time Per Scene | RealEstate10K | | | DyCheck | | | NVIDIA | | |
|---|---|---|---|---|---|---|---|---|---|---|
| | | ↑PSNR | ↑SSIM | ↓LPIPS | ↑mPSNR | ↑mSSIM | ↓mLPIPS | ↑PSNR | ↑SSIM | ↓LPIPS |
| *Static Feed-forward* | | | | | | | | | | |
| DepthSplat (Xu et al., 2025a) | 0.60s | 26.57 | 80.06 | 0.124 | 13.83 | 0.4364 | 0.3850 | 17.16 | 0.5002 | 0.3023 |
| GS-LRM† (Zhang et al., 2024) | 0.57s | 26.94 | 79.13 | 0.139 | 14.60 | 0.4535 | 0.3775 | 17.83 | 0.4988 | 0.3265 |
| Ours (static) | 0.84s | **27.60** | 81.25 | 0.113 | 15.24 | 0.4784 | 0.3783 | 18.73 | 0.5042 | 0.2959 |
| *Optimization-based* | | | | | | | | | | |
| Splatter-a-Video (Sun et al., 2024) | 37min | - | - | - | 13.61 | 0.3131 | 0.5706 | 14.39 | 0.2538 | 0.5983 |
| Shape-of-Motion (Wang et al., 2025b) | 10min | - | - | - | 17.96 | 0.5662 | 0.3463 | 15.30 | 0.3169 | 0.5087 |
| MoSca (Lei et al., 2025) | 45min | - | - | - | 18.24 | 0.5514 | 0.3698 | **21.45** | **0.7123** | **0.2653** |
| Ours | **0.93s** | 26.98 | 81.75 | 0.111 | **18.46** | **0.5887** | **0.3094** | 19.16 | 0.5141 | 0.3152 |

used to unproject tracked points into 3D space. To account for scale differences across methods, we normalize all predicted 3D points by their median norm before evaluation.

Quantitative results are reported in Table 3. While 3D-based SpatialTracker generally outperforms 2D-based approaches, all of them rely heavily on pretrained monocular depth estimators for geometry reasoning, introducing significant noise and inconsistency in the 3D space. In contrast, MOVIES directly estimates 3D point positions in a *shared* world coordinate, enabling more accurate and robust 3D tracking, and achieves consistently superior or competitive performance across all datasets. Visualization results are provided in Figure 7 in the appendix and the supplementary material.

## 4.4 ABLATION AND ANALYSIS

**Camera conditioning**  We investigate different camera conditioning strategies in the static pertaining stage. Quantitative comparisons are provided in Table 4. Camera tokens are injected throughout the feature backbone, enabling effective camera-aware modeling. In contrast, Plücker embeddings provide limited conditioning on their own and are merely comparable to having no camera information at all. However, as a pixel-aligned representation, Plücker embeddings are complementary to camera tokens, and their combination yields the most effective camera conditioning.

**Motion Supervision**  To learn object 3D movements in dynamic scenes, we provide two kinds of motion supervision: (1) point-wise L1 loss and (2) distribution loss, as described in Equation 4. Their

Table 3: **Evaluation on 3D Point Tracking**. The best , second best and third best results are highlighted for clarity. † denotes combining a depth estimation model (Chen et al., 2025a).

| 3D Point Tracking | Aria Digital Twin | | | DriveTrack | | | Panoptic Studio | | |
|---|---|---|---|---|---|---|---|---|---|
| | $\downarrow$ EPE$_{3D}$ | $\uparrow \delta_{3D}^{0.05}$ | $\uparrow \delta_{3D}^{0.10}$ | $\downarrow$ EPE$_{3D}$ | $\uparrow \delta_{3D}^{0.05}$ | $\uparrow \delta_{3D}^{0.10}$ | $\downarrow$ EPE$_{3D}$ | $\uparrow \delta_{3D}^{0.05}$ | $\uparrow \delta_{3D}^{0.10}$ |
| BootsTAPIR (Doersch et al., 2024)† | 0.5539 | 17.73% | 32.97% | 0.0617 | 55.82% | 75.66% | 0.0650 | 69.28% | 87.95% |
| CoTracker3 (Karaev et al., 2024)† | 0.5614 | 19.88% | 35.82% | 0.0637 | 55.30% | 77.55% | 0.0617 | 69.27% | 88.04% |
| SpatialTracker (Xiao et al., 2024) | 0.5413 | 18.08% | 38.23% | 0.0648 | 56.58% | 80.67% | 0.0519 | 72.91% | 89.86% |
| Ours | 0.2153 | 52.05% | 71.63% | 0.0472 | 60.63% | 79.87% | 0.0352 | 87.88% | 94.61% |

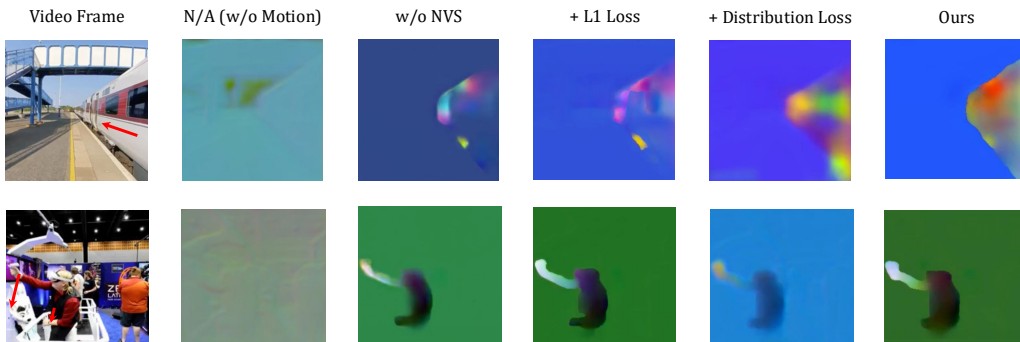

Figure 4: **Motion Visualization for Ablation Studies**. We investigate key factors affecting motion learning in MOVIES, such as loss design and synergy with view synthesis. XYZ values in motion maps are normalized as RGB for visualization. Red arrows on video frames indicate motion directions.

efficiency is evaluated via the 3D point tracking task in Table 5. Without any motion supervision, i.e., learning solely from novel view synthesis, training exhibits severe loss oscillations and frequent None gradients. The distribution loss captures only relative motion between pixels, while point-wise L1 loss produces more reasonable motion maps. Combining both leads to sharper boundaries. Qualitative results of estimated motions from different motion objectives are provided in Figure 4.

**Synergy of Motion and View Synthesis** Thanks to the unified design of MOVIES, it supports simultaneous novel view synthesis (NVS) and motion estimation. We study their mutual benefits in Table 6. "**NVS w/o motion**" disables explicit motion supervision during training, relying solely on NVS as a proxy to learn dynamics. As shown in Table 6 and Figure 4, this setting fails to learn meaningful motion and tends to model static scenes. "**Motion w/o NVS**" detaches the motion head from 3DGS rendering and instead conditions the depth head on time. Although explicit supervision enables some motion learning, the predictions are blurry and low-quality, as shown in Figure 4. Moreover, the depth head must now model both geometry and dynamics, increasing its burden and negatively affecting NVS. These results highlight the mutual reinforcement between NVS and motion estimation in MOVIES, where joint training leads to better performance on both.

### 4.5 ZERO-SHOT APPLICATIONS

**Scene Flow Estimation** Scene flow can be naturally derived by transforming the estimated per-pixel motion vectors from world coordinates to the target camera's coordinates. Visualization results in Figure 5(a) present sharp edges and accurate motion directions. Different colors in the flow maps indicate different movement directions that are marked by block arrows in the figure. More visualization results are provided in Figure 8 in the appendix and the supplementary material.

**Moving Object Segmentation** By thresholding the norm of per-pixel motion vectors, the estimated motion maps can also be used to segment moving objects (Figure 5(b)), which is an essential task in computer vision and robotics (Xie et al., 2024; Huang et al., 2025). Remarkably, this is achieved without any explicit supervision during training, demonstrating the strong potential of our method. More visualization results are provided in Figure 9 in the appendix and the supplementary material.

Table 4: **Ablation study** on camera conditioning.

| Camera Conditioning | RealEstate10K | | |
|---|---|---|---|
| | ↑ PSNR | ↑ SSIM | ↓ LPIPS |
| N/A | 25.56 | 74.13 | 0.150 |
| + Plücker embedding | 25.81 | 74.44 | 0.143 |
| + Camera token | _26.81_ | _78.65_ | _0.121_ |
| Ours (static) | **27.60** | **81.25** | **0.113** |

Table 5: **Ablation study** on motion supervision.

| Motion Supervision | Aria Digital Twin | | |
|---|---|---|---|
| | ↓ $\text{EPE}_{3D}$ | ↑ $\delta_{3D}^{0.05}$ | ↑ $\delta_{3D}^{0.10}$ |
| N/A | 0.7938 | 19.58% | 32.86% |
| + Point-wise L1 | _0.2262_ | _48.74%_ | _69.93%_ |
| + Distribution loss | 0.2496 | 45.98% | 66.87% |
| Ours | **0.2153** | **52.05%** | **71.63%** |

Table 6: **Ablation Study** on the synergy of motion estimation and novel view synthesis (NVS).

| Motion & View Synthesis | DyCheck | | | NVIDIA | | | Aria Digital Twin | | |
|---|---|---|---|---|---|---|---|---|---|
| | ↑ mPSNR | ↑ mSSIM | ↓ mLPIPS | ↑ PSNR | ↑ SSIM | ↓ LPIPS | ↓ $\text{EPE}_{3D}$ | ↑ $\delta_{3D}^{0.05}$ | ↑ $\delta_{3D}^{0.10}$ |
| NVS w/o motion | 15.82 | _0.4796_ | _0.3741_ | 18.38 | 0.4856 | **0.3010** | 0.7938 | 19.58% | 32.86% |
| Motion w/o NVS | _16.26_ | 0.4556 | 0.3461 | _18.98_ | _0.4939_ | 0.3207 | _0.3801_ | _24.72%_ | _42.92%_ |
| Ours | **18.46** | **0.5887** | **0.3094** | **19.16** | **0.5041** | _0.3152_ | **0.2153** | **52.05%** | **71.63%** |

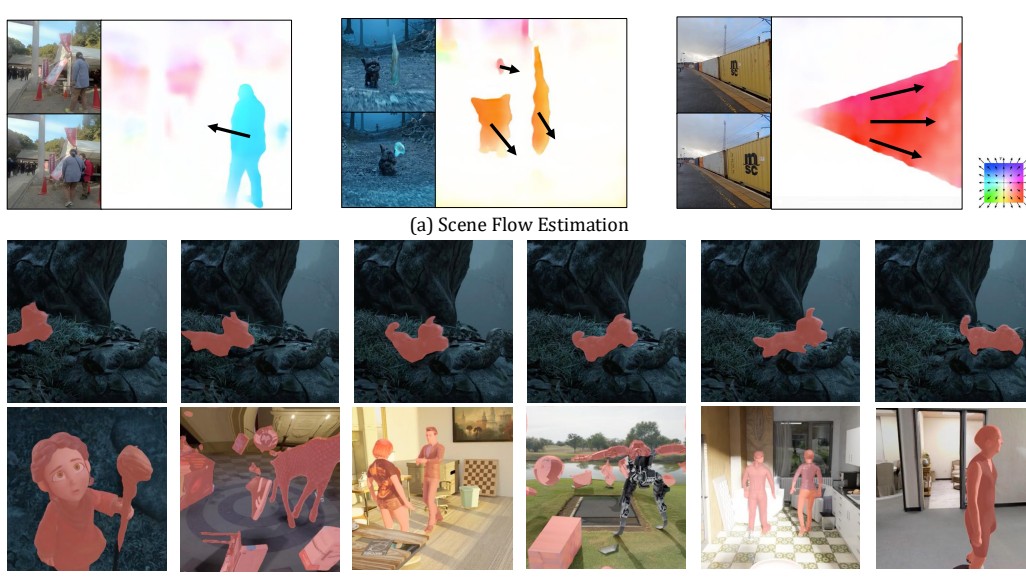

(a) Scene Flow Estimation

(b) Moving Object Segmentation

Figure 5: **Zero-shot Applications**. The predicted pixel-aligned motion maps from our model can be directly applied to downstream tasks, such as (a) scene flow estimation and (b) moving object segmentation, in a **zero-shot** manner, without any task-specific fine-tuning or supervision.

## 5 CONCLUSION

We introduce MOVIES, a feed-forward model for dynamic novel view synthesis from monocular videos. It's trained on large-scale datasets from diverse sources and jointly models scene appearance, geometry and motion in a unified and efficient network. Dynamic splatter pixels are proposed to represent dynamic scenes, enabling accurate and temporally coherent 4D reconstruction. Beyond novel view synthesis, MOVIES supports a broad range of applications including depth estimation, 3D point tracking, scene flow estimation and moving object segmentation, demonstrating its versatility for dynamic scene perception. We hope this work could serve as a step toward generalizable dynamic scene understanding, and support applications that require spatial and motion intelligence. Discussions on limitations and future work are included in Appendix B.

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

# A    IMPLEMENTATION DETAILS

## A.1    DATASET DETAILS

As summarized in Table 1, the majority of training datasets used in our experiments are synthetic, providing rich annotations such as pixel-aligned depth maps and accurate camera intrinsics and extrinsics. Ground-truth 3D point trajectories are also available in PointOdyssey (Zheng et al., 2023) and DynamicReplica (Karaev et al., 2023). Stereo4D (Jin et al., 2025), on the other hand, is a large-scale dataset constructed from YouTube stereo videos and annotated using pretrained foundation models (Teed & Deng, 2020; Doersch et al., 2024). Despite being partially noisy, its diversity and scale offer strong generalization benefits.

The pretrained backbone, VGGT (Wang et al., 2025a), requires all 3D scenes to be normalized to a unit scale, which in turn necessitates depth maps and camera extrinsics to be defined in a consistent metric space. It is satisfied by the synthetic datasets and Stereo4D, whose camera poses and depths are metric-aligned by construction. However, RealEstate10K (Zhou et al., 2018) only provides relative camera parameters estimated via COLMAP (Schonberger & Frahm, 2016), resulting in an unknown global scale. To address this issue, we re-estimate both depth maps and camera extrinsics using recent foundation models, including Video Depth Anything (Chen et al., 2025a), Depth Pro (Bochkovskii et al., 2024) and MegaSaM (Li et al., 2025), to recover aligned geometry across frames.

## A.2    CURRICULUM TRAINING

We observed that training MOVIES is particularly unstable: the training loss often fluctuates abruptly, and gradients are prone to becoming `None`. It may arise from sparse annotations and the heterogeneous nature of the training data, which mixes datasets from various sources with differing domains (e.g., indoor vs. outdoor, real vs. synthetic), camera fields of view, recording frame rates, etc.

Thanks to the versatile design of MOVIES, we employ a curriculum strategy that gradually increases the training complexity. It begins by pretraining the model on low-resolution ($224 \times 224$) static datasets with only depth and photometric losses, then introduces dynamic datasets along with motion supervision. We found that static datasets play a crucial role in stabilizing training for dynamic scenes, without which the loss would be highly unstable. Since modeling dynamic scenes requires reconstructing a set of 3DGS for each query time, which results in high GPU memory usage, we start by training on 5 input views for dynamic scenes and then expand to 13 views for fine-tuning. Finally, the training resolution is increased to $518$. Similar to VGGT (Wang et al., 2025a), in the last training stage, we randomly sample the frame number from $2 \sim 13$ and the aspect ratio from $0.5 \sim 2$, with the largest side is fixed to $518$.

## A.3    TRAINING DETAILS

Image encoder, feature backbone and depth head of MOVIES are initialized from VGGT (Wang et al., 2025a). Motion head is initialized from its point head. Remaining components, such as the splatter head and camera/time embeddings, are trained from scratch. We use AdamW optimizer (Loshchilov & Hutter, 2018) with a weight decay of $0.05$, and adopt a cosine learning rate scheduler (Loshchilov & Hutter, 2016) with linear warm-up for all curriculum training stages. For static pretraining and dynamic scenes with 5 and 13 input views at a resolution of $224 \times 224$, we use learning rates of $4e-4$, $4e-4$ and $4e-5$, respectively, with a batch size of $256$. Training rates for parameters initialized from VGGT are multiplied by $0.1$. With $32$ H20 GPUs, training takes about $2$ days for static and then dynamic scenes with 5 views, and $2$ days for 13 views. MOVIES is then finetuned on $518 \times 518$ videos with 13 frames using a learning rate of $1e-5$, which takes around $1$ day. To improve memory and computation efficiency, several techniques such as `gsplat` (Ye et al., 2025b) rendering backend, DeepSpeed (Rasley et al., 2020), gradient checkpointing (Chen et al., 2016), gradient accumulation, and `bf16` mixed precision are adopted in this work.

For the multi-task training objective in Equation 2 and 4, We did not manually tune the relative weights between different loss terms. Instead, the weights were set to bring the numerical values of the losses into roughly similar ranges: $\lambda_d = \lambda_r = \lambda_{pt} = 1$ and $\lambda_m = \lambda_{dist} = 10$.

## B LIMITATIONS AND FUTURE WORK

Although MOVIES achieves competitive performance with inference speeds that are orders of magnitude faster than optimization-based methods, there remains a noticeable gap in reconstruction quality. This gap arises partly because many optimization-based approaches benefit from multiple pretrained models that preprocess input videos to provide richer and more accurate cues. Incorporating such richer prior knowledge directly into MOVIES represents a promising direction for future work to further improve reconstruction fidelity and robustness.

Currently, MOVIES depends on off-the-shelf tools for camera parameter estimation, which adds an external dependency and may limit end-to-end optimization. Seamlessly integrating camera pose estimation within the MOVIES pipeline could simplify the overall workflow, reduce error accumulation, and enhance adaptability to diverse scenarios.

Moreover, scaling MOVIES to handle long videos and achieve high-resolution rendering remains a challenge. The computational cost and memory demand grow significantly with scene complexity and resolution, which constrains practical deployment. Developing more compact and efficient dynamic scene representations, possibly through novel model architectures or sparse encoding strategies, is essential to push 4D reconstruction towards real-world applications.

## C USE OF LARGE LANGUAGE MODELS

We disclose that LLMs were used exclusively for language editing and polishing purposes in this work. Their role was limited to improving clarity, readability and grammar in the manuscript. LLMs were not used for generating research ideas, designing methods, analyzing data, conducting experiments, or writing substantive technical content. All conceptual contributions, experiments, analyses and conclusions are solely the work of the authors.

## D LICENSE INFORMATION

We employ several open-source implementations in our experimental comparisons, including: (1) DepthSplat (Xu et al., 2025a)[1] (MIT License), (2) Splatter-a-Video (Sun et al., 2024)[2] (Apache License), (3) Shape-of-Motion (Wang et al., 2025b)[3] (MIT License), (4) MoSca (Lei et al., 2025)[4] (MIT License), (5) BootsTAPIR (Doersch et al., 2024)[5] (Apache License), (6) CoTracker3 (Karaev et al., 2024)[6] (Creative Commons Attribution-NonCommercial 4.0 International Public License), and (7) SpatialTracker (Li et al., 2024)[7] (Attribution-NonCommercial 4.0 International).

Datasets from diverse sources are utilized to train MOVIES, including: (1) RealEstate10K (Zhou et al., 2018)[8] (Creative Commons Attribution 4.0 International License), (2) TartanAir (Wang et al., 2020)[9] (Creative Commons Attribution 4.0 International License), (3) MatrixCity (Li et al., 2023a)[10] (Apache License), (4) PointOdyssey (Zheng et al., 2023)[11] (MIT License), (5) DynamicReplica Karaev et al. (2023)[12] (Attribution-NonCommercial 4.0 International License), (6) Spring (Mehl et al., 2023)[13] (CC BY 4.0), (7) VKITTI2 (Cabon et al., 2020)[14] (Creative Commons Attribution-NonCommercial-ShareAlike 3.0), and (8) Stereo4D (Jin et al., 2025)[15] (CC0 1.0 Universal).

---

[1] https://github.com/cvg/depthsplat/tree/main
[2] https://github.com/SunYangtian/Splatter_A_Video
[3] https://github.com/vye16/shape-of-motion/
[4] https://github.com/JiahuiLei/MoSca
[5] https://github.com/google-deepmind/tapnet
[6] https://github.com/facebookresearch/co-tracker
[7] https://github.com/henry123-boy/SpaTracker
[8] https://google.github.io/realestate10k/
[9] https://theairlab.org/tartanair-dataset/
[10] https://city-super.github.io/matrixcity/
[11] https://pointodyssey.com/
[12] https://github.com/facebookresearch/dynamic_stereo
[13] https://spring-benchmark.org/
[14] https://europe.naverlabs.com/research/computer-vision/proxy-virtual-worlds-vkitti-2/
[15] https://stereo4d.github.io/

## E  MORE VISUALIZATION RESULTS

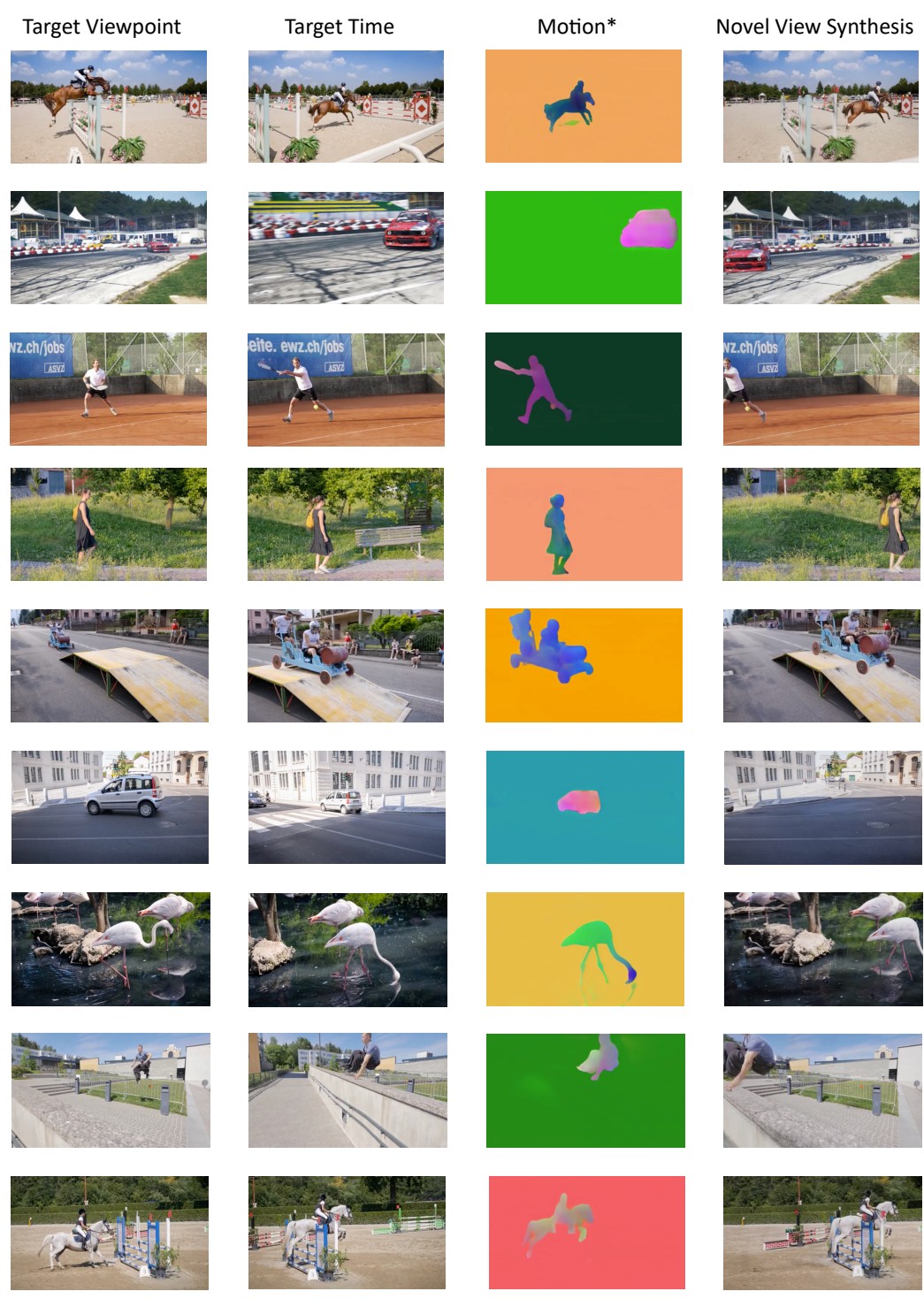

Figure 6: Qualitative results of novel view synthesis on the DAVIS dataset (Perazzi et al., 2016). "Motion*" denotes the 3D pixel displacements between the "Target Time" frame with respect to the "Target Viewpoint" frame. The "Novel View Synthesis" results show the reconstructed scene at the target time from the target viewpoint.

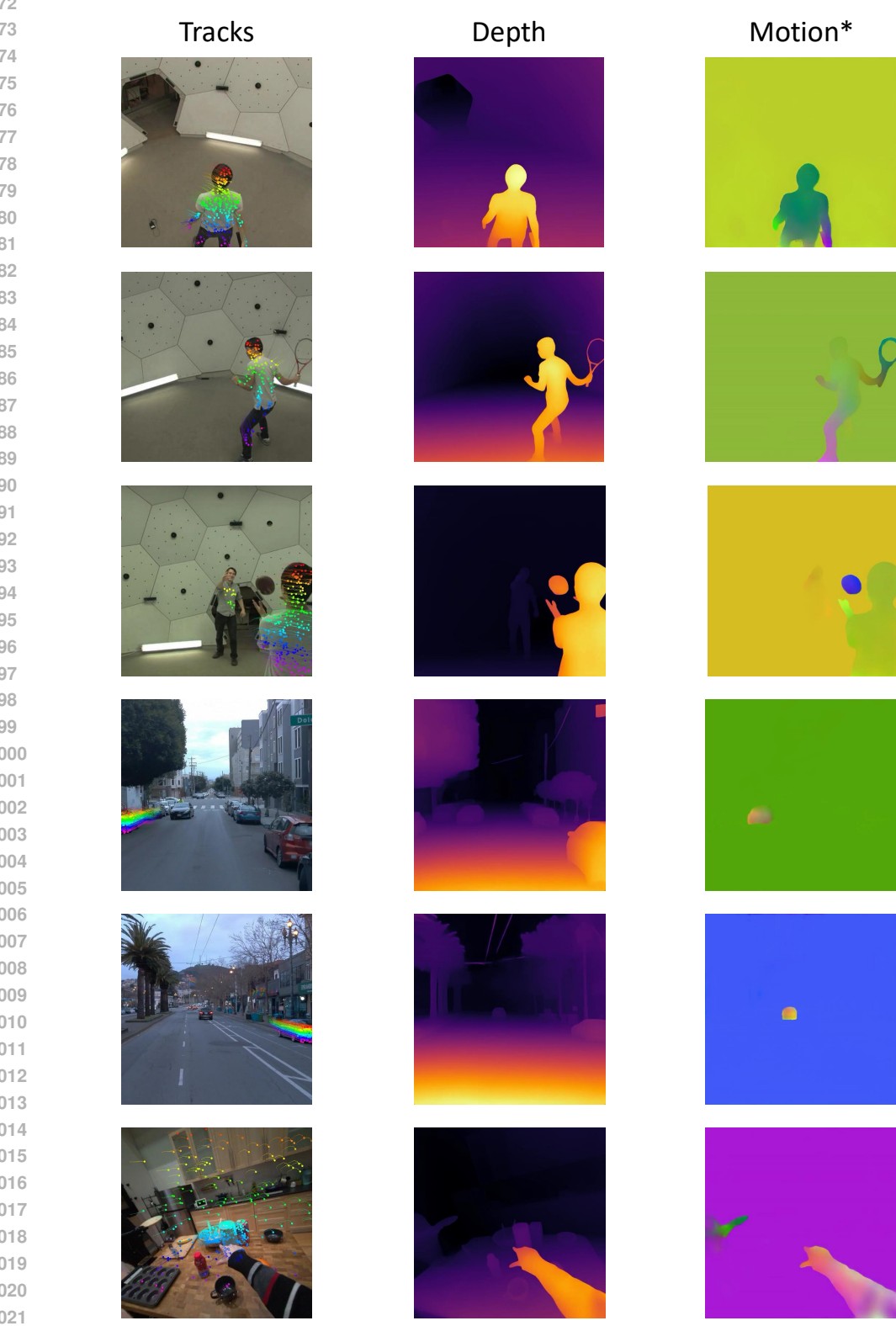

Figure 7: Qualitative results for 3D point tracking. "Motion*" means the 3D points' movements of the input frame with respect to the first one.

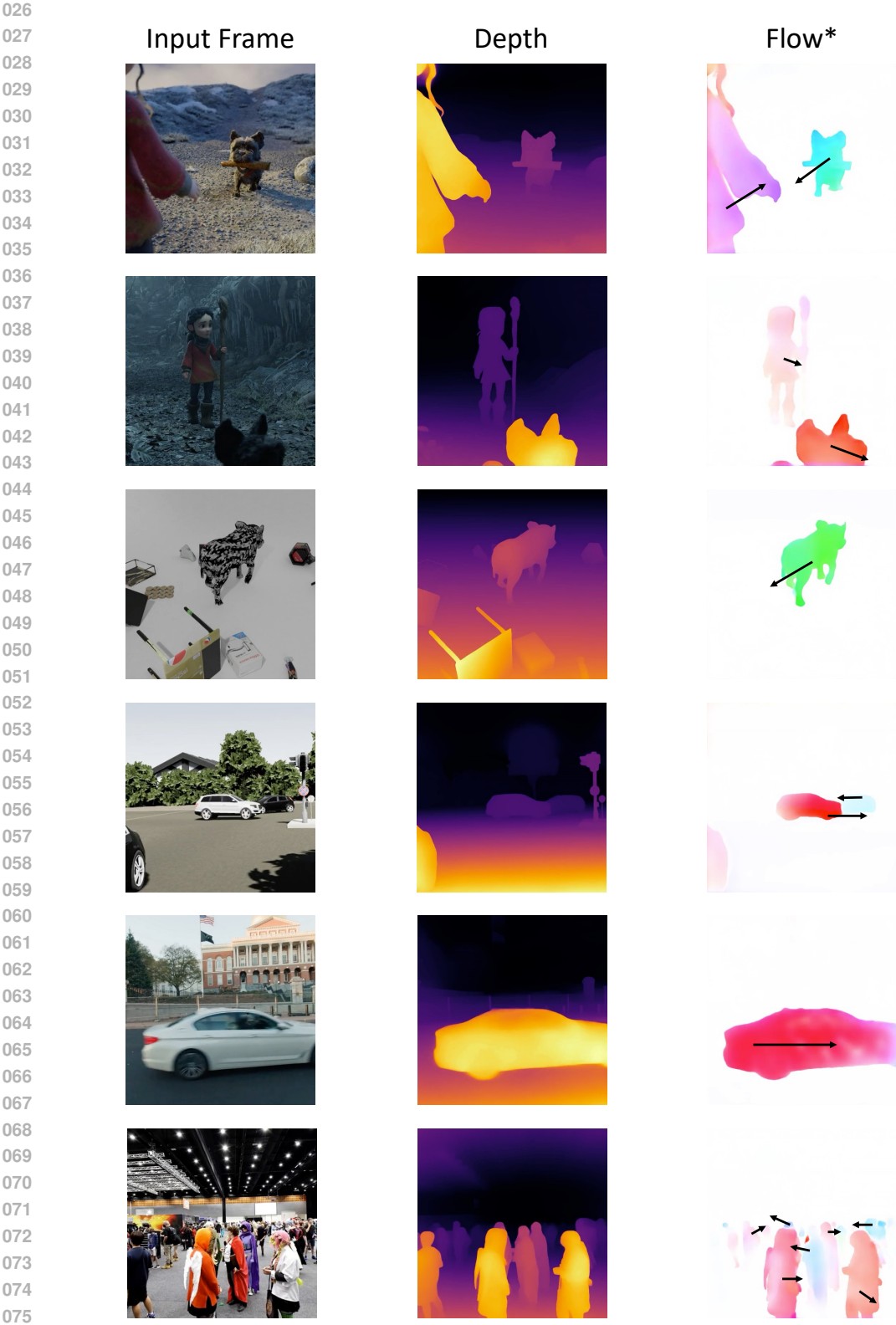

Figure 8: More results for scene flow estimation. "Flow*" means the optical flow of the input pixels with respect to the first frame. Arrows on pictures roughly mark directions.

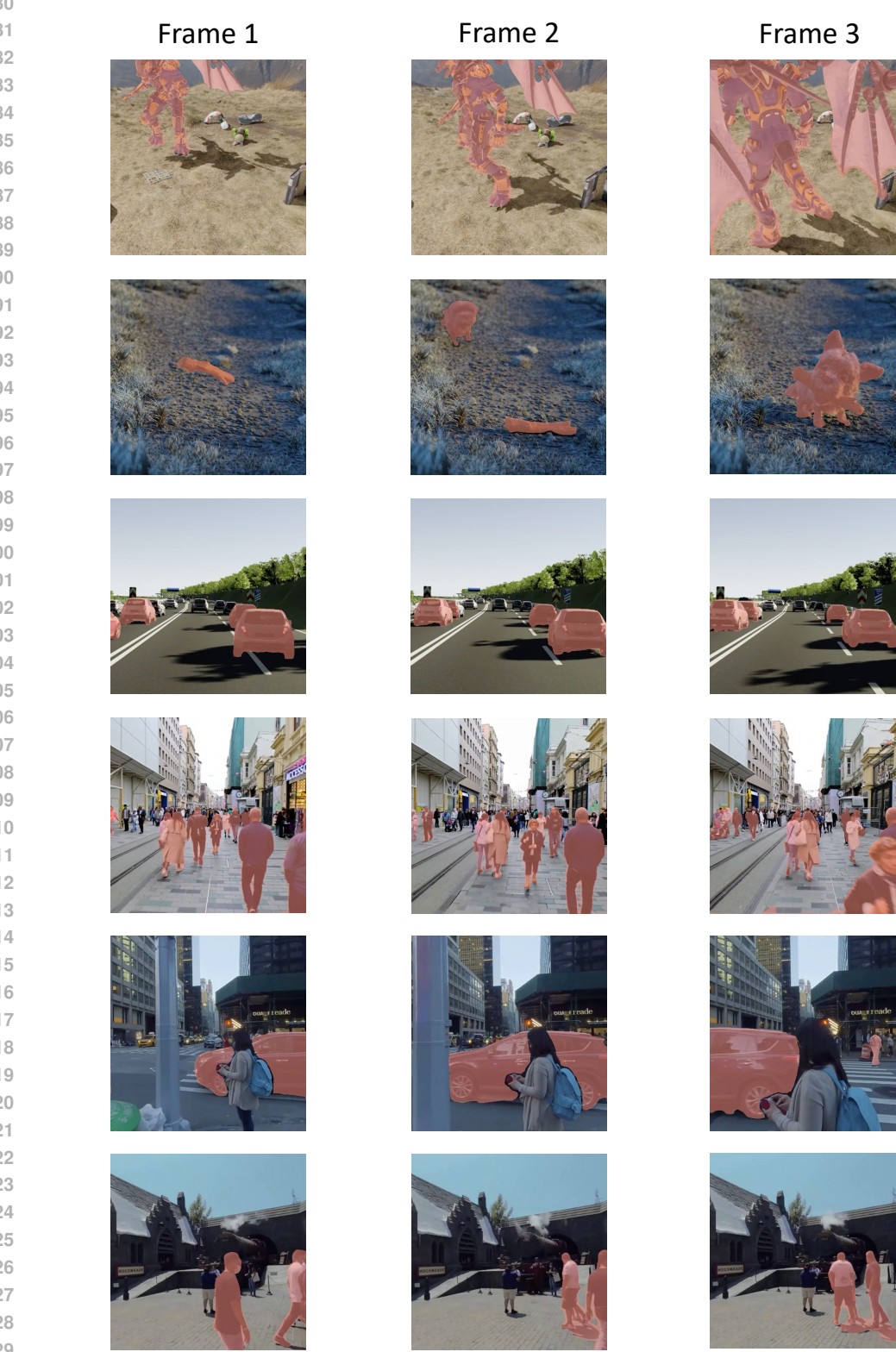

Figure 9: More results for moving object segmentation. Masks for moving parts, which are filtered from motion maps, are highlighted across frames in light red.

