# OpenReview forum: "MoVieS: Motion-Aware 4D Dynamic View Synthesis in One Second"
_ICLR.cc/2026/Conference — ICLR 2026 Conference Withdrawn Submission_

### Official Review · Reviewer_Va4v · 2025-10-20

**Soundness:** 3
**Presentation:** 3
**Contribution:** 2
**Rating:** 2
**Confidence:** 5

**Summary:**

MOVIES is a **feed-forward framework** that generates **4D dynamic novel views** from **monocular videos in just one second**. It models dynamic 3D scenes through **pixel-aligned Gaussian primitives** and explicitly learns their **temporal motions**, allowing for a unified representation of **appearance, geometry, and motion** within a single network.

By **integrating view synthesis and geometry reconstruction**, MOVIES enables **large-scale training** across diverse datasets with minimal reliance on task-specific supervision. This unified design also facilitates **zero-shot capabilities**, including **scene flow estimation** and **moving object segmentation**. Extensive experiments show that MOVIES delivers **competitive results** across multiple tasks while achieving **significant speed improvements** during inference.

**Strengths:**

**[S1]** The model introduces minimal architectural modifications to VGGT while achieving impressive performance. By naturally extending VGGT, it demonstrates strong **novel view synthesis (NVS)** results across various datasets. The **depth** and **splat heads** directly predict **3D Gaussian Splatting (3D-GS)** representations, enabling **real-time multi-view rendering**. The **motion head** predicts **time-conditioned motion fields**, effectively modeling **dynamic splatter pixels**—a novel approach to dense motion prediction.

**[S2]** The paper provides **comprehensive ablation studies** to validate the effectiveness of each model component. Through well-controlled experiments, the authors clearly demonstrate how each design choice contributes to overall performance.

**[S3]** The **writing is clear and well-organized**, with concepts and figures effectively illustrating the core ideas. The presentation's clarity makes the methods and experimental purposes easy to follow. I have included a few **minor comments** on this in the **Weakness Section**.

**Weaknesses:**

[W1] **Assumption of pose-awareness**: their strong assumption of pose-awareness within the underlying scene severely limits the applicability of the model in real-world scenarios. Given that VGGT jointly estimates camera parameters along with other dense outputs such as depth, pointmaps, and tracking features, I am curious why the authors decided to exclude camera tokens from their model architecture. Since MoVieS is primarily trained on datasets where camera parameters are available (e.g., RealEstate10K and Stereo4D), it would be beneficial to relax or explicitly examine the pose-awareness assumption. This is critical since despite efficiently handling videos with poses, they still require an additional process to obtain the camera poses, which takes a long time and relies on heavy computations. For instance, if we desire to use traditional SfM, COLMAP, it would even take longer than previous pose-free per-scene optimizatation approaches such as RoDynRF[2], RoDyGS[3], and SoM[4]. Therefore, they should also provide comparison with the pose-free approaches in terms of reconstruction accuracy and overall optimization time (including pose estimation).

Some wrong claims:

(L303-305): Although they claimed their model does not require heavy pretrained models and complex multi-stage pipelines, they should rely on heavy SfM pipelines or pretrained models for pose estimation.

[W2] **Missing benchmarks**. They missed several recent work in this research area. In DyCheck, they miss the comparison with D-NeRF, 4DGS, and Deform3D which are 3~4PSNR above their method. Moreover, they have compared their model with pose-free approaches (SoM, MoSca), which are toally unfair comparison. Note that SoM and MoSca do not adopt known camera poses and instead rely on initialized poses from other fast SfM tools. By adding such baselines, MoVieS shows notably worse performance than optimization-based methods. Therefore, the authors should tone-down their claim for clarity.

[W3] **An unified DPT head for depth and Gaussian attributes**: it is unclear why the authors chose to separate the DPT heads for depth and Gaussian attributes. From an implementation perspective, I understand that recent deep learning frameworks make it easier to implement separate DPT modules for predicting depth maps and splatter-related attributes. However, conceptually, predicting per-pixel Gaussian attributes should either follow the prediction of the depth head or explicitly refer to the geometric information provided by it, since they are strongly correlated for reliable novel view rendering. The paper would be clearer if the authors compared their model with a variant in which the DPT and splat heads are merged—i.e., a single DPT head jointly predicts both depth and splat attributes.

[W4] **Missing illustration of predicted motions**. Since they introduce a novel representation for motion, they should show that their motions are clearly drawn in 3D space. To show that I suggest to add a figure like MoSca Figure 1 to draw the trajectory over time. Authors cannot undertand how the motion is predicted based on Figure 6,7,8,9.

[W5] **Quantitative evaluation of motions in static scene**s. One of the issues of using dynamic models on static scenes is the ambiguity between camera movements and object movements. Since the authors claimed that their model achieves natural convergence to zero for static scenes (L294-L296), they should quantitatively show that their model

[W6] **Explanation of Plucker embedding (L177 - L179)**. The term *“pixel-aligned”* is not incorrect; however, readers who are not familiar with Plücker embeddings might find this expression confusing. I recommend explicitly stating that the ray corresponding to each pixel is converted into a Plücker coordinate. Additionally, it would be helpful to include a simple equation illustrating how each camera embedding is formulated.

[W7] **Missing information in Fig 1.** In Fig. 1, the assumption under pose-awareness is not clearly illustrated. Although the authors describe it by drawing `Camera Pos.' in the figure, it has potential for misunderstanding since the model's input seem like unposed videos. It would be clearer if they clearly illustrate that both videos and their frame-wise camera poses are given as inputs to the model.

[W8] **Not self-contained writing**. Academic paper should be self-contained, in other words, all details should be elaborated within this paper. I understand that the page limit often leads to the omissions of unimportant concepts. However, it should be at least provided in the Appendix chapter to make writing more clear. The authors should supplement their writing with the listed components below:

- The concept of confidence-based weighting on training objectives.
- The exact form of rendering rendering loss (with their respective weight value $lambda).

**Questions:**

- Why do  the authors use a different scale of SSIM between Table2 (RealEstate 10K and NVIDIA)? For RealEstate10K, they used 100-scales; however, for NVIDIA, they used a normalized scale to 1.
- Did they fully fine-tuned VGGT? or did they simply froze the layers? It is unclear for me to fine-tune VGGT since it no more uses camera tokens from VGGT, which makes attention operates differently. Could you clarify the details for this?

---

### Official Review · Reviewer_dYmc · 2025-11-01

**Soundness:** 3
**Presentation:** 2
**Contribution:** 3
**Rating:** 6
**Confidence:** 4

**Summary:**

This paper presents MoVieS, a feed-forward model for 4D dynamic scene reconstruction from monocular videos. The key contribution is the introduction of dynamic splatter pixels, which decouple 3D Gaussian primitives from the time-dependent deformation field, enabling modelling of geometry and motion within a single framework. The model achieves competitive performance on novel view synthesis and 3D point tracking tasks while providing large speedups compared to per-scene optimisation.

**Strengths:**

- To my knowledge, this is the first work to address dynamic reconstruction in a feed-forward manner (at least to satisfactory quality).
- The dynamic splatter representation is intuitive and theoretically sound.
- The work makes good use of a strong VGGT prior.
- The training regime with complementary supervision from multiple datasets is an interesting way of making use of incomplete data.
- The qualitative results look visually impressive.
- Similarly, the quantitative evaluation shows good performance with fast inference speed.
- The auxiliary zero-shot applications are very useful as well.

**Weaknesses:**

- With high computational cost, the adoption of the framework most likely depends on the code and pretrained model release.
- The model is initialised with a VGGT backbone, which raises the question of how much of the performance comes from VGGT pretraining (would be a useful ablation).
- It seems that training is very sensitive and needs a heavily engineered curriculum.
- It would be good to perform a deeper analysis on motion supervision, e.g. 2000x oversampling on Spring seems quite strong.
- It is a bit unclear whether the performance comes from algorithmic contribution or the scale of the training.

**Questions:**

- It would be good to know which datasets are crucial for the training, and what scale is required (or is it diversity that matters).
- The paper describes the limitations of the approach. I wonder if authors can comment on the most common failure modes of the approach when run on in-the-wild videos.

---

### Official Review · Reviewer_4wg9 · 2025-11-01

**Soundness:** 3
**Presentation:** 3
**Contribution:** 3
**Rating:** 4
**Confidence:** 5

**Summary:**

The paper introduces a 4D dynamic view synthesis method.
The methods inputs a sequence of frames and their poses, and a transformer-based architecture outputs 3D points, 3D Gaussian attributes, and 3D motion in a feed-forward manner. Given the estimates the method can solve several tasks such as novel-view synthesis and 3D point tracking.

---

The paper's contribution is great: a feedforward 4D dynamic view synthesis model that is competitive to the optimization-based approaches (Table 2). However, there are many unclear points (eg., possible unfair comparison, unclear motion color coding, Figure 2 visualization, inaccurate motion estimation, generalization, motion segmentation, etc) that makes hard to give an accept recommendation confidently. For now, my decision is between **2: reject and 4: borderline reject** (but would like to increase based on responses.)

**Strengths:**

* **Easy to follow**

  The paper is easy to follow. It is easy to understand the technical details.

* **Good ablation study**

  Especially Figure 4 and Table 6 show ablation study on the loss function. It validates the effectiveness of the proposed ideas on motion loss, NVS loss, L1 loss, and distribution loss.

* **Good empirical results on benchmark datasets**

  In Table 2, the method shows better accuracy than other feed-forward NVS methods (eg., DepthSplat, GS-LRM) or optimization-based methods with order of magnitude faster runtime. This can be considered as a good empirical contribution of the paper.

**Weaknesses:**

* **Possible unfair comparison in Table 3**

  The other methods (BootsTAPIR, CoTracker3, and SpatialTracker) possibly don't use camera pose input for the inference. Given that the proposed method uses precomputed (or given) camera pose, I wonder how fair the comparison would be. It would be curious to know i) how the evaluation is actually conducted where the other methods don't have pose information, ii) what pose estimator (or GT pose?) the proposed method uses.


* **Unclear motion color coding**

  The color coding of the motion can be misleading. Not only moving objects, background also seems to have non-zero motion (ie., non-white color coding), as in all Figures, except Figure 5. Does it mean that the background usually have constant 3D motion? Depending on the answers, it could change the understanding of the coordinate representation, eg., if 3D motion is defined at the global camera coordinate or local camera coordinate.

* **Figure 2: 3D GS or points?**

  I was wondering if the RGB visualization in Figure 2 can be misleading. At a glance, it doesn't look like a visualization of 3D GS (where each Gaussian have different 3D scale, opacity, etc.) but a set of uniform dilated 3D points that are usually seen from feedforward 3D point estimation methods (eg., VGGT, DUSt3R, MonST3R).

* **Accumulating all 3D Gaussians from all views?**

  When rendering images, I wonder if the method accumulates all the 3D Gaussians from all input frames and rasterizes images. I wonder if this occurs any memory problem or computational burden. If it accumulates all 3D Gaussians from all views, I wonder if it is a redundant design that a single 3D point is represented by multiple overlapped moving 3D Gaussians from multiple views instead of a single 3D Gaussian that moves across the multiple frames.

* **Inaccurate motion?**

  In the motion visualization in Figure 4 (upper, the most right figure), Figure 7 (bottom), some motion visualization includes non-homogeneous motion in foreground objects that exhibits rigid motion (although the depth visualization looks correct). I am wondering what's happening in the motion space.

* **Generalization for the ablation study**

  Some ablation study (Table 4 and Table 5) is done on a single dataset (RE10K or ADT) for different modality evaluations. I was wondering if the paper can provide a ablation study on multiple datasets. For example, it would be possible to report the camera conditioning and motion supervision analyses on both RE10K and ADT together. This can further prove the effectiveness of the proposed ideas.

  Is there a specific reason why Table 4 evaluates Ours (**static**) instead of Ours?

* **Motion object segmentation**

  For Fig 5 (b), the motion object segmentation is tested on the training domain (Spring, PointOdyssey). It would be curious to see how the method actually performs in-the-wild examples.

**Questions:**

* **Sensitivity to the camera pose input**

  The reliance on the camera pose can be a limitation, as stated in the limitation section. It would be curious how robust the method is to the noisy camera input. One could evaluate the method on a benchmark dataset by injecting random noise to the camera pose input at different levels.

* At **Line 303**

  How many images does each scene include? Because depending on the number of input frames, the runtime of the method could change, and each scene can include different number of frames.

* **static**?

  What does the *static* mean in Table 2? (eg., *Static* feed-forward and Ours (*static*))

---

### Official Review · Reviewer_LYRv · 2025-11-01

**Soundness:** 3
**Presentation:** 3
**Contribution:** 3
**Rating:** 6
**Confidence:** 4

**Summary:**

The paper introduces MOVIES, a feed-forward model for dynamic 4D scene reconstruction and new view synthesis from monocular videos. It jointly models appearance, geometry, and motion using 3D Gaussian Splatting (3DGS), enabling efficient depth estimation, 3D point tracking, scene flow estimation, and moving object segmentation.

**Strengths:**

1. Efficient and Unified Approach: MOVIES integrates appearance, geometry, and motion using 3D Gaussian Splatting (3DGS), enabling fast inference while achieving competitive performance across multiple benchmarks like RealEstate10K and TAPVid-3D.

2. Zero-Shot Capabilities: The model demonstrates strong zero-shot performance in tasks such as scene flow estimation and moving object segmentation, showcasing its versatility and potential for real-world applications.

**Weaknesses:**

1. Artifacts in Video Results:
The video results show artifacts, such as blurred legs and embedded wheels, impacting visual coherence. The authors should provide a detailed discussion on the causes (e.g., model limitations or data issues) to better understand the model's performance and limitations.

2. Camera Pose Prediction:
The removal of camera pose prediction, despite VGGT’s ability to estimate it, is unclear. Since the model still requires camera pose estimation, the authors should explain why they chose to exclude the VGGT camera head.

3. Main Modification Compared to VGGT:
MOVIES replaces VGGT’s prediction head with 3D Gaussian Splatting (3DGS), but this change seems to offer limited novelty or significant modification to the model.

4. Ablation Study on Camera Conditioning:
The ablation study on camera conditioning lacks meaningful insight. With explicit camera conditioning and additional input information, it is expected that the model’s performance will naturally improve, making the results less informative.

**Questions:**

See the weakness part

---

### Note · Authors · 2025-11-12

I have read and agree with the venue's withdrawal policy on behalf of myself and my co-authors.